# Olfactory Impairment Is the Main Predictor of Higher Scores at REM Sleep Behavior Disorder (RBD) Screening Questionnaire in Parkinson’s Disease Patients

**DOI:** 10.3390/brainsci13040599

**Published:** 2023-03-31

**Authors:** Paolo Solla, Qian Wang, Claudia Frau, Valentina Floris, Francesco Loy, Leonardo Antonio Sechi, Carla Masala

**Affiliations:** 1Neurological Unit, AOU Sassari, University of Sassari, Viale S. Pietro 10, 07100 Sassari, Italy; 2Department of Biomedical Sciences, University of Sassari, Viale S. Pietro 10, 07100 Sassari, Italy; 3Department of Biomedical Sciences, University of Cagliari, SP 8 Cittadella Universitaria, 09042 Monserrato, Italy

**Keywords:** Parkinson’s disease, olfactory dysfunction, REM sleep behavior disorder

## Abstract

Introduction: Olfactory impairment and REM sleep behavior disorder (RBD) are common non-motor symptoms in Parkinson’s disease (PD) patients, often preceding the onset of the specific motor symptoms and, thus, crucial for strategies directed to anticipate PD diagnosis. In this context, the specific interaction between olfactory impairment and RBD has not been clearly defined. Objective: The aim of this study was to determine the possible role of olfactory impairment and other clinical characteristics as possible predictors of higher scores at RBD screening questionnaire (RBDSQ) in a large population of PD patients. Methods: In this study, 590 PD patients were included from the Parkinson’s Progression Markers Initiative. Demographic and clinical features were registered. All participants completed motor and non-motor evaluations at the baseline visit. For motor assessments, the disease severity was evaluated by the Movement Disorder Society-Unified Parkinson’s Disease Rating Scale (MDS-UPDRS) pars III. Regarding non-motor symptoms assessment, Montreal Cognitive Assessments (MoCA), University of Pennsylvania Smell Identification Test (UPSIT) and RBD screening questionnaire (RBDSQ) were registered. Results: Among 590 PD patients included in this study, 111 patients with possible RBD were found (18.8%). RBD was less frequent in female PD patients (*p*  ≤  0.011). Among patients with or without possible RBD diagnosis, statistically significant differences in MDS-UPDRS III (23.3 ± 11.4 vs. 19.7 ± 9.1, respectively, *p*  ≤  0.002) and in UPSIT score (19.7 ± 8.3 vs. 22.6 ± 8.0, respectively, *p*  ≤  0.001) were found. Moreover, significant correlations between RBDSQ versus UPDRS III score and versus UPSIT score were observed. Multivariate linear regression analysis showed that UPSIT was the most significant predictor of higher scores at RBDSQ, while the other significant predictors were UPDRS III and age. Conclusions: The severity of olfactory impairment appears tightly correlated to RBD symptoms, highlighting the role of these biomarkers for PD patients. Additionally, according to this large study, our data confirmed that RBD in PD patients exhibits peculiar gender differences.

## 1. Introduction

Parkinson’s disease is the second most common neurodegenerative disease after Alzheimer’s disease and is characterized by motor symptoms (such as resting tremor, rigidity, postural instability, and bradykinesia), and non-motor symptoms including olfactory dysfunctions, sleep disorders, cognitive impairment, autonomic dysregulation, etc. [1]. Among non-motor symptoms, olfactory impairment represents the most common disturbance affecting 96% of PD patients, often preceding the onset of the specific motor symptoms [2,3], and deteriorating more rapidly in the early phase of the disease [4] or in association with non-tremor dominant PD subtypes [5,6,7]. Olfactory impairment as a prodromal non-motor symptom is crucial for strategies directed to anticipate the diagnosis of PD, bearing in mind that motor symptoms appear only after a loss of a conspicuous number of neurons due to nigral degeneration with consequent striatal dopamine depletion [8]. In this process of progressive neurodegeneration, Braak and colleagues have identified a peculiar spreading pattern of aggregated and misfolded α-synuclein proteins, which are the major constituents of Lewy bodies, starting in the caudal brainstem and advancing rostrally through the upper brainstem, limbic regions, and finally the neocortex [9]. The prodromal phase of PD corresponds to a disease stage where neurodegeneration processes, described as Braak stages 1–3, involve extranigral sites, such as the olfactory bulb and tracts, the lower brainstem, and the peripheral autonomic nervous system [9]. Interestingly, regarding the impact on the olfactory structures of the neuroinflammatory processes in PD and neurodegenerative diseases, it would be valuable to elaborate on the neuroinflammatory pathogenesis [10,11].

Moreover, in addition to olfactory disturbances, other non-motor symptoms in PD patients such as REM sleep behavior disorder (RBD), constipation, and depression may precede the movement disorders for years [12]. These symptoms are often associated, and the relationship between olfactory impairment and other non-motor symptoms such as depression and constipation is well-known [13,14]. Among them, RBD has been defined as a complex multidimensional parasomnia characterized by dream enactment and complex motor behaviors during REM sleep, which may cause injury and loss of muscle atonia during REM sleep, known as REM sleep without atonia [15]. This peculiar sleep disorder is strongly associated with synucleinopathy neurodegeneration, as most patients with RBD develop signs and symptoms relative to a synucleinopathy, such as PD, Lewy body disease or multiple system atrophy within approximately 10 years [16,17]. Indeed, there are converging proofs substantiating that idiopathic RBD is often a prodromal form of synucleinopathy [16]. These findings include neurodegenerative biomarker research, longitudinal cohort outcome investigations, and pathological confirmation from both autopsy series and demonstration of extranigral α-synuclein pathology in living patients with RBD [16]. Furthermore, a recent review has highlighted that RBD, which has been primarily described in synucleinopathies, has also been found to be present in other atypical parkinsonisms such as progressive supranuclear palsy syndrome (PSPS) and corticobasal syndrome (CBS) [18]. Among RBD patients, it was observed that 74% of subjects met Movement Disorders Society criteria for a diagnosis of prodromal PD [19]. Moreover, the presence of a probable RBD was associated with a faster progression of motor symptoms in PD patients with a postural instability and gait dysfunction (PIGD) phenotype in comparison with those presenting with a tremor dominant subtype [20].

Identification of prodromal symptoms constitutes an even greater unmet need given that new potential disease-modifying drugs will have their greatest possibility for success at these initial manifestations. In this context, the identification of olfactory disorders and RBD as early as possible would probably be useful both in clinical practice and in clinical trials of potentially neuroprotective treatments.

While olfactory impairment may be assessed and diagnosed using validated clinical instruments such as the University of Pennsylvania Smell Identification Test (UPSIT) [21] and the Sniffin’ Sticks Extended test [22], establishing a RBD diagnosis commonly required a polysomnography (PSG), which represents the gold standard [23]. However, in studies evaluating a large population of patients, the RBD screening questionnaire represents a validated diagnostic instrument consisting of a 10-item patient self-rating questionnaire (maximum total score: 13 points) covering the clinical features of RBD [24] and representing a useful tool for the screening of RBD in PD patients [25]. In this context, the specific interaction between motor and non-motor symptoms, such as olfactory impairment and RBD in PD patients have not been evaluated, especially in a large population [25,26,27]. In fact, the assessment of olfactory impairment, and especially odor identification, may help to predict the onset of a Lewy body disease in patients with idiopathic RBD over a relatively short time period [28]. Furthermore, in RBD patients, anosmia predicts a higher short-term risk of transition to Lewy body disease, although it cannot distinguish between PD and Lewy body disease [29]. In the current study, we aim to determine the potential role of olfactory impairment and other factors (age at the onset, disease duration, sex, cognitive abilities, and motor impairment) as possible predictors of higher scores at the RBD screening questionnaire (RBDSQ) in a large population of PD patients.

## 2. Materials and Methods

### 2.1. Study Population

A total of 590 PD patients (235 women and 355 men) with a mean age of 61.4 ± 10.1 years were included in this study from Parkinson’s Progression Markers Initiative (PPMI). PPMI is an ongoing comprehensive observational, multicenter, international project started in June 2010 and designed to identify biomarkers relative to PD progression. Indeed, the primary aim of PPMI is to identify genomic, biochemical, or imaging biomarkers of clinical progression finalized both to improve the understanding of PD etiology and course and to provide crucial tools to enhance the likelihood of success of PD modifying therapeutic trials [30,31]. PPMI data are publicly available (www.ppmi-info.org/data), accessed on 9 January 2023, and updated in real-time, and rely on a partnership of government, PD foundations, academics and industry working cooperatively. The detailed study protocol, manuals, and storage processes are available at www.ppmi-info.org/study-design, accessed on 9 January 2023. This project has been approved by the institutional review board at each site, with written informed consent obtained from all subjects before enrolment of PD patients in the project. The study was performed in agreement with relevant guidelines and regulations.

According to PPMI protocol, inclusion criteria for PD patients were: (1) at least two of the following signs: bradykinesia, resting tremor, and rigidity or an asymmetric bradykinesia or asymmetric resting tremor; (2) diagnosis for 2 years or less; (3) Hoehn and Yahr stage I or II at observation; (4) striatal dopamine transporter deficits on (123I)IFP-CIT SPECT imaging, greater in the contralateral to the most clinically affected side; (5) PD drug naivety; and (6) male or female age 30 years or older at time of PD diagnosis. Exclusion criteria were the following: (1) receiving any of the following drugs: neuroleptics, alpha methyldopa, methylphenidate, reserpine, metoclopramide, or amphetamine derivative, within 6 months of age at observation; and (2) use of investigational drugs or devices within 60 days prior to observation (dietary supplements taken outside of a clinical trial are not exclusionary, e.g., coenzyme Q10).

### 2.2. Assessments

Demographic and clinical features were registered. All participants completed motor and non-motor evaluations at the baseline visit. For motor assessments, disease severity was evaluated by the Movement Disorder Society-Unified Parkinson’s Disease Rating Scale (MDS-UPDRS) part III [32].

Regarding non-motor symptoms, cognitive abilities were assessed using the Montreal Cognitive Assessments (MoCA) [33]. The used version of the MoCA is a one-page 30-point test (available at www.mocatest.org) that evaluates different domains (visual-constructional skills, executive functions, attention and concentration, memory, language, conceptual thinking, calculations, and spatial orientation) and should be administered approximately in ten minutes. The total score of MoCA is 30, and any score ≥26 is considered as normal [33].

Instead, olfactory function was evaluated using UPSIT, a smell identification test with 40 items at a suprathreshold level, which provides an indication of quantitative smell loss as anosmia and different levels of hyposmia. The UPSIT test consists of 40 questions with 4 diverse 10-page booklets. The number of items out of 40 that were perceived correctly served as the dependent measure [21]. As previously described in a systematic review, PD patients were defined as anosmic if their score was ≤18/40, whereas they were considered normosmic if their scores were >33 in men or >34 in women [34].

The REM sleep behavior disorder was registered using RBD screening questionnaire (RBDSQ) [24]. This questionnaire consists of 10 items and is a patient self-rating instrument assessing the subject’s sleep. The bed partner’s input may be useful but not necessary, because patients do not always have a long-time companion. The maximum total score of the RBDSQ is 13 points [24].

In agreement with a previous study revealing the best cut-off value for RBD using the RBDSQ [25], a score superior to 6 points was set as the cut-off value for a possible RBD diagnosis.

### 2.3. Statistical Analysis

The SPSS software, version 26 (IBM Corporation, Armonk, NY, USA), was used for statistical analysis. All data were presented as mean values ± standard deviation (SD). Variables normality was checked with the Kolmogorov–Smirnov test. For comparison of the demographic and clinical features between PD with or without diagnosis of possible RBD, all variables were assessed by means of independent sample *t* test or the Yates-corrected chi-square test, as appropriate.

Bivariate correlations among demographic data, clinical, motor, and non-motor assessments were performed using Pearson’s correlation coefficient (r). Moreover, a multivariate linear regression analysis was performed to assess the potential contribution of each significant variable on RBD severity in PD patients. The multivariate linear regression analysis was performed using age at observation, gender, PD duration, MDS-UPDRS III, UPSIT score and MoCA score as independent variables, while RBDSQ score was considered the dependent variable. A *p* value < 0.05 was considered statistically significant.

## 3. Results

Demographic and clinical features of all PD patients, also differentiated in patients with or without possible RBD diagnosis, were reported in Table 1.

Among 590 PD patients included in this study, 235 were women (39.8%), while patients with possible RBD were 111 (18.8%). PD patients with or without possible RBD diagnosis were similar for age and PD duration, with mean age at observation equal to 61.4  ± 10.1 years, while mean PD duration was 2.7 (SD 3.3) years. Prevalence on female PD patients with possible RBD was minor in comparison with those without RBD (28.8% versus 42.3%, respectively; *p*  ≤  0.011). Among patients with or without possible RBD diagnosis, statistically significant differences in MDS-UPDRS III (23.3 ± 11.4 vs. 19.7 ± 9.1, respectively; *p*  ≤  0.002) and in UPSIT score (19.7 ± 8.3 vs. 22.6 ± 8.0, respectively; *p*  ≤  0.001) were found. PD patients with or without RBD did not show a significant impairment in MoCA scores (26.7 ± 2.9 vs. 26.8 ± 2.8, respectively).

At the olfactory assessment, among 590 PD patients, 213 were defined as anosmic (36%), whereas 62 patients were considered as normosmic. Among PD patients with anosmia, 67 (28.5%) were women and 146 (41.5%) were men. Thus, a gender difference in olfactory impairment was found, with the prevalence on PD patients affected by anosmia significantly lower in the female gender in comparison with males (chi-square with Yates correction: 9.2169; *p*  ≤ 0.0025).

In Table 2, significant correlations among demographic data, clinical, motor, and non-motor assessments in patients affected by PD were reported.

In particular, we found statistically significant correlations between RBDSQ versus UPDRS-III score (r = 0.150, *p* < 0.001) and versus UPSIT score (r = −0.179, *p* < 0.001). Other statistically significant correlations were found between UPSIT score versus age at observation (r = 0.204, *p* < 0.001), versus UPDRS-III score (r = 0.180, *p* < 0.001), and versus MoCA score (r = 0.121, *p* < 0.003).

Furthermore, to better clarify the impact of bivariate correlations related to RBDSQ score, a multivariate linear regression analysis was performed to predict higher scores at RBDSQ in PD patients in relation to demographic/clinical data (age at observation, sex, disease duration), motor (UPDRS III) and non-motor symptoms (UPSIT and MoCA) as independent variables (Table 3). In the multivariate linear regression analyses, the RBDSQ was set as a dependent variable, while age at observation, sex, PD duration, UPDRS-III, UPSIT score, and MoCA score were independent variables.

Multivariate linear regression analysis showed that the mean value of the UPSIT test (Figure 1A), which indicated the olfactory function of the patients, was the most significant predictor for higher scores in the RBDSQ [F_(6,589)_ = 6.634, *p* < 0.0001], while other significant predictors were UPDRS-III (Figure 1B) and age (*p* < 0.001 and *p* < 0.003, respectively) (Figure 1C). The model explained around 25% of the variance (R^2^ = 0.253).

## 4. Discussion

The aim of this study was to determine the potential role of olfactory dysfunction and other factors (such as age at the onset, sex, cognitive abilities, and motor symptoms) as potential predictors of higher scores at the RBDSQ in PD patients, investigating a large population of PD patients. RBD is tightly associated with PD and other synucleinopathies such as dementia with Lewy bodies and multiple system atrophy, often starting before the onset of any clear specific signs or symptoms of these neurodegenerative diseases, proposing that RBD may be one of the most important prodromal manifestations [16].

From a general point of view, we found a frequency of possible RBD near to 19% in all enrolled PD patients. This finding appears in line with other previous studies identifying a presumptive prevalence of RBD in 15–40% of PD patients using RBD screening scales, quite different from the frequency of confirmed RBD in up to 62.5% in studies using video polysomnography [25,35,36,37]. At the same time, we found that a large group of parkinsonian patients from the same population presented an evident olfactory disorder (anosmia) with frequency higher than 35%, confirming anosmia as a well-documented marker of PD [2,3]. Thus, comparing RBD and smell disturbances, it emerges that the presence of anosmia in PD patients was more frequent than RBD. This finding is line with previous studies suggesting that olfactory dysfunction is reported in 96% of PD patients [2,3].

Interestingly, we also found that patients with possible RBD presented with a greater impairment at the motor assessment with MDS-UPDRS-III. This finding appears to be in agreement with previous studies reporting that RBD presence in individuals with PD was associated with a severe clinical form of the disease [36] documenting a risk factor for PD forms with greater severity of motor symptoms [35].

However, the main finding of our study was the demonstration that olfaction impairment and RBD were strongly associated. In this context, we found that the decrease of the UPSIT score was the most significant predictor of higher scores at RBDSQ, while other predictors such as UPDRS-III and age at observation were less significant. Thus, according to our data, the presence of a more severe olfactory impairment is strongly correlated with a more symptomatic expression of this specific parasomnia.

A probable reason of this association between RBD and impaired olfactory function is likely due to the proximity of the specific brainstem nuclei involved in these pathologies, as well documented by the anatomopathological studies of Braak and colleagues [9].

Indeed, it is well-known that physiological sleep atonia during REM sleep is modulated by the subcoeruleus nucleus, together with the amygdala [38]. The main pathways of these structures are involved in the inhibitory inputs to the neurons of the spinal anterior horns, determining the loss of skeletal muscular tone [39,40]. In this scenario, the loss of physiological atonia during sleep in PD patients with RBD is related to the reduction of signal intensity in the locus coeruleus/subcoeruleus [38]. Moreover, structural and functional neuroimaging abnormalities in the brainstem, and more precisely with involvement in the dorsal pons, likely impacts REM sleep atonia control [41]. It is interesting to notice that damage in the forebrain cholinergic system is considered a strong candidate for explaining the marked olfactory impairment among neurological disorders, often with significant destruction of the nucleus basalis of Meynert or ascending basolateral cholinergic circuits [42].

In this regard, previous investigations have suggested an association between RBD and olfactory impairment [26,43,44]. For example, Stiasny-Kolster et al. found that almost all the RBD patients (97%) showed an impairment of the olfactory threshold, while 63% of these patients had lost the skill to differentiate odors [43]. However, in our opinion, the present study provides more defined evidence due to the large number of PD patients included.

Interestingly, we observed a significant gender difference in PD patients with possible RBD, with more men affected by this peculiar sleep disorder in comparisons with women. In this regard, it is remarkable to note that previous research has shown conflicting results. Indeed, although previous reports in RBD suggested the presence of a gender difference [45,46], one of the most important epidemiological investigations carried out in a middle-to-older age population-based sample did not show any difference between men and women [47]. The findings of our study, which has the advantage of having been conducted in a large population of parkinsonian patients, indicate the effective presence of a gender difference in the frequency of this sleep disorder, with a greater impact on male gender.

In fact, gender differences are not uncommon in PD, particularly among the wide spectrum of non-motor symptoms [48,49], and especially in a specific prodromal disturbance such as olfactory impairment [50]. Furthermore, the present study also confirmed the presence of gender differences regarding the olfactory impairment, with male PD patients more anosmic when compared to females, supporting the hypothesis that men may show an increased risk to develop these non-motor symptoms.

This evidence strongly suggests that future investigations on PD patients should be more focused on sex differences since the necessity to develop gender-tailored management in PD is more and more evident. In this context, the gender may provide insights into mechanisms of neurodegeneration and improve epidemiologic and therapeutic clinical trial designs. In fact, the possibility to focus PD patients on specific subtypes appears as a key resource for drug development and understanding of the clinical and biological features of PD progression milestones. Moreover, the full definition of gender-related differences may play an important role to improve the precision medicine approach in PD.

Regarding this issue, for example, there is also a lack of data on sex differences in response to antiparkinsonian drugs and adverse events. The same issue is widely underestimated in the investigations related to the prodromal symptoms of PD, and the present study provides interesting data in this regard. Forthcoming studies are therefore needed to elaborate on gender-tailored management in PD [51].

As a limitation, it should be noted that the study presented with a cross-sectional design, without investigating possible correlations over time and measure longitudinal changes. Another limitation was the lack of REM sleep confirmation by video PSG, which is mandatory for a definite RBD diagnosis. However, since this study was designed as an investigation in a large population, PSG was not performed because the examination is time- and labor-consuming, while an appropriate questionnaire for RBD screening in clinical settings was used. In this context, the use of RBDSQ in this large population appears reasonable for the screening of this complex parasomnia, and this study should be considered as a pilot study. As a further limitation, we did not explore the relationship of olfactory impairment and RBD in less severe tremor-dominant PD compared to the severe PIGD motor subtype. We retain that future investigations focused on this relationship may give further insight into disease mechanisms.

## 5. Conclusions

In conclusion, this study showed that the presence of olfactory dysfunction appears to be tightly correlated with RBD symptoms, highlighting the role of these biomarkers for PD patients. Additionally, according to this large study, our data confirmed that RBD in PD patients exhibits peculiar gender differences. The evaluation of olfactory dysfunction and RBD may play a key role both in clinical practice and in the application of potentially neuroprotective treatments in PD patients. Further studies are required to confirm the presence and progression of this relationship between these PD biomarkers also using video-PSG and with longitudinal follow-up.

## Figures and Tables

**Figure 1 brainsci-13-00599-f001:**
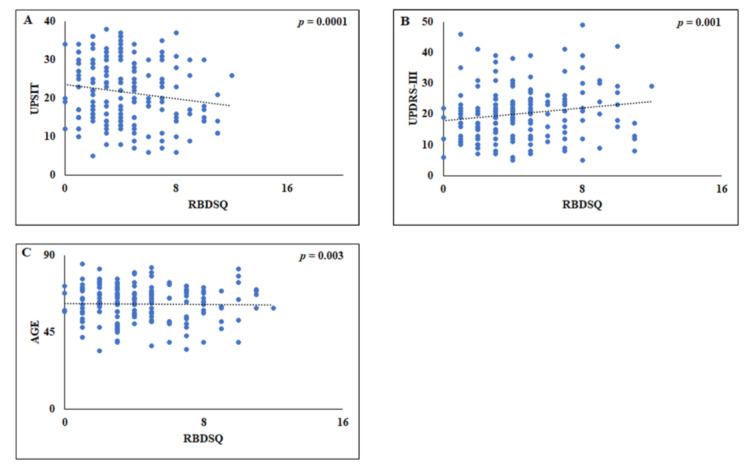
Scatterplots of the relationship between Rapid Eye Movement Sleep Behavior Disorder Screening Questionnaire (RBDSQ) versus University of Pennsylvania Smell Identification Test (UPSIT) (**A**), between RBDSQ versus Unified Parkinson’s disease rating scale part III (UPDRS) (**B**), and between RBDSQ versus age (**C**).

**Table 1 brainsci-13-00599-t001:** Demographic and clinical information of PD patients.

	Total PD Patients	PD Patients without RBD	PD Patients with Possible RBD	*p* Value
Demographics	*n* = 590	*n* = 479 (81.2%)	*n* = 111 (18.8%)	
Age at observation	61.4 (10.1)	61.7 (10.1)	60.1 (10.2)	0.135
PD duration (years)	2.7 (3.3)	2.7 (3.5)	2.7 (2.2)	0.948
Sex *n*, female (%)	235 (39.8%)	203 (42.3%)	32 (28.8%)	**0.011**
MDS-UPDRS-III	20.4 (9.7)	19.7(9.1)	23.3 (11.4)	**0.002**
RBDSQ	4.1 (2.8)	3.0 (1.6)	8.7 (1.6)	**0.0001**
UPSIT	22.0 (8.1)	22.6 (8.0)	19.7 (8.3)	**0.001**
MoCA	26.7 (2.9)	26.8 (2.8)	26.7 (2.9)	**0.387**

PD = Parkinson’s disease; SD = standard deviation; MDS-UPDRS-III = Movement Disorder Society Revision of the Unified Parkinson’s disease rating scale part III; RBDSQ = Rapid Eye Movement Sleep Behavior Disorder Screening Questionnaire; *n* = number; UPSIT = University of Pennsylvania Smell Identification Test; MoCA = Montreal Cognitive Assessment. Significant *p* values are highlighted in bold. Data are expressed as mean (SD) or mean (percentage).

**Table 2 brainsci-13-00599-t002:** Correlations among demographic data, clinical, motor, and non-motor assessments in patients affected by Parkinson’s disease.

		RBDSQ	Age	UPDRS-III	MoCA	UPSIT	PD Duration
RBDSQ	r	1	−0.072	0.150	−0.004	−0.179	−0.006
	*p*		0.081	**0.001**	0.921	**0.001**	0.883
Age	r	−0.072	1	0.129	−0.189	−0.204	0.023
	*p*	0.081		0.002	0.001	0.001	0.577
UPDRS-III	r	0.150	0.129	1	−0.147	−0.180	0.183
	*p*	**0.001**	**0.002**		**0.001**	**0.001**	**0.001**
MoCA	r	−0.004	−0.189	−0.147	1	0.121	−0.086
	*p*	0.921	**0.001**	**0.001**		**0.003**	**0.037**
UPSIT	r	−0.179	−0.204	−0.180	0.121	1	−0.016
	*p*	**0.001**	**0.001**	**0.001**	**0.003**		0.702
PD duration	r	−0.006	0.023	0.183	−0.086	−0.016	1
	*p*	0.883	0.577	**0.001**	**0.037**	0.702	

RBDSQ = Rapid Eye Movement Sleep Behavior Disorder Screening Questionnaire; UPDRS-III = Movement Disorder Society Revision of the Unified Parkinson’s disease rating scale part III; MoCA = Montreal Cognitive Assessment; UPSIT = University of Pennsylvania Smell Identification Test; PD = Parkinson’s disease; r = Pearson’s correlation coefficient. Significant *p* values are highlighted in bold.

**Table 3 brainsci-13-00599-t003:** Multivariate linear regression analyses using RBDSQ as a dependent variable and age at observation, gender, PD duration, UPDRS-III, UPSIT score and MoCA score as independent variables.

	Unstandardized Coefficients		Standardized Coefficients		
	B	Std Error	β	*t*	*p*
(Costant)	6.276	1.455		4.314	0.000
UPSIT	−0.060	0.014	−0.175	−4.183	**0.0001**
UPDRS-III	0.039	0.012	0.137	3.269	**0.001**
Age	−0.034	0.011	−0.124	−2.974	**0.003**
Gender	0.221	0.232	0.039	0.955	0.340
MoCA	0.012	0.039	0.012	0.294	0.769
PD duration	−0.021	0.035	−0.025	−0.603	0.547

RBDSQ = Rapid Eye Movement Sleep Behavior Disorder Screening Questionnaire; UPDRS-III = Unified Parkinson’s disease rating scale part III; MoCA = Montreal Cognitive Assessment; UPSIT = University of Pennsylvania Smell Identification Test; PD = Parkinson’ s disease; Std Error: Standard error of the mean. Significant *p* values are highlighted in bold.

## Data Availability

Not applicable.

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
