# Peer review of "Olfactory Impairment Is the Main Predictor of Higher Scores at REM Sleep Behavior Disorder (RBD) Screening Questionnaire in Parkinson’s Disease Patients"

_brainsci, 2023, doi:10.3390/brainsci13040599_

Round 1
Reviewer 1 Report
Comments and Suggestions for Authors
Dear authors
Thank you for submitting your article. Here are my comments:
1. The sequence of non-motor presentations of PD is important and if possible, it would benefit the article to emphasize on the sequence of occurrence of RBD and olfactory dysfunction. This could lead to earlier detection of the disease
2. To my knowledge, there are no effective disease modifying drugs for PD and other synucleinopathies which undermines the values of the study to some extent, but I can see the promising path that these studies provide.
Author Response
- The sequence of non-motor presentations of PD is important and if possible, it would benefit the article to emphasize on the sequence of occurrence of RBD and olfactory dysfunction. This could lead to earlier detection of the disease.
We strongly agree with the Reviewer consideration that emphasizing on the sequence of occurrence of RBD and olfactory dysfunction may improve the article, because the sequence of non-motor presentations of PD could lead to earlier detection of the disease. In this regard, we have added this consideration in the manuscript and this new reference has been added.
Mahlknecht P, Iranzo A, Högl B, Frauscher B, Müller C, Santamaría J, Tolosa E, Serradell M, Mitterling T, Gschliesser V, Goebel G, Brugger F, Scherfler C, Poewe W, Seppi K; Sleep Innsbruck Barcelona Group. Olfactory dysfunction predicts early transition to a Lewy body disease in idiopathic RBD. Neurology. 2015 Feb 17;84(7):654-8.
Assessment of olfactory function, particularly odor identification, may help to predict the development of a Lewy body disease in patients with iRBD over a relatively short time period and thus to identify patients suitable for future disease modification trials.
- To my knowledge, there are no effective disease modifying drugs for PD and other synucleinopathies which undermines the values of the study to some extent, but I can see the promising path that these studies provide.
We agree with the Reviewer that, at the moment, there are not effective disease modifying drugs for PD and other synucleinopathies, however the knowledge of prodromal symptoms may be useful to identify patients suitable for future disease modification trials.
Reviewer 2 Report
Comments and Suggestions for Authors
Title: The title is too long and may be modified.
Language: Major grammatical corrections must be made in the abstract and throughout the manuscript.
Abstract: In the results, UPDRS III and UPSIT score differences whether greater or lesser between patients with or without possible RBD diagnosis terms is to be mentioned.
Introduction:
Both RBD and association with both PIGD and Tremor dominant PD may be mentioned.
The relationship of olfactory impairment to non-motor symptoms like depression, constipation may be added.
Is RBD associated with olfactory disturbances in other synucleinopathies? The authors may provide references.
Materials and Methods:
The methods have to be curtailed. Appropriate references of the validated scales and methods are sufficient. For eg. details of various domains for MoCA, UPSIT and details of RBDSQ may be omitted.
Inclusion and exclusion criteria are vague.
Results:
Difference in gender for olfaction impairment is not clear. Language editing is required.
Fig 1. Scatter plots may be improved by increasing the size.
Discussion:
RBD as the most important functional risk marker for prodromal PD is now known. Numerous studies have also evaluated the association between olfactory dysfunction and motor and non-motor symptoms in PD. Over here, the authors show that olfaction impairment and RBD to be strongly associated as their main finding and that anosmia is more frequent than RBD in PD patients. The gender difference is also highlighted.
There are certain concerns regarding the paper.
Firstly, questionnaires to screen for RBD are less specific and sensitive than video-PSG, the gold standard for testing RBD. However, it is understood that for such a large population of patients, PSG was not feasible.
Secondly, exploring the relationship of olfactory impairment and RBD in less severe tremor-dominant PD compared to the severe PIGD motor subtype may give further insight into disease mechanisms. Longitudinal design of the study would have added more strength to the present findings
The last paragraph of the discussion can be eliminated.
Major language editing is needed in the revised paper.
Author Response
Title: The title is too long and may be modified.
According to reviewer suggestion, the title has been shortened.
Language: Major grammatical corrections must be made in the abstract and throughout the manuscript.
According to Reviewer suggestion, major grammatical corrections have been made and are indicated on the text. Moreover, the manuscript has now been revised by a native speaker.
Abstract: In the results, UPDRS III and UPSIT score differences whether greater or lesser between patients with or without possible RBD diagnosis terms is to be mentioned.
According to Reviewer suggestion, UPDRS III and UPSIT score differences between patients with or without possible RBD diagnosis have been included in the abstract.
Introduction:
Both RBD and association with both PIGD and Tremor dominant PD may be mentioned.
According to Reviewer suggestion, the association between RBD and PIGD/Tremor dominant PD has been included in the Introduction with the following sentence: “Moreover, the presence of a probable RBD was associated with a faster progression of motor symptoms in PD patients with a postural instability and gait dysfunction (PIGD) phenotype in comparison with those presenting with a tremor dominant subtype [19].” Moreover, the following reference has been added in the list: Duarte Folle A, Paul KC, Bronstein JM, Keener AM, Ritz B. Clinical progression in Parkinson's disease with features of REM sleep behavior disorder: A population-based longitudinal study. Parkinsonism Relat Disord. 2019 May;62:105-111.
The relationship of olfactory impairment to non-motor symptoms like depression, constipation may be added.
According to Reviewer suggestion, the following sentence is now included on the text “These symptoms are often associated, and the relationship between olfactory impairment and other non-motor symptoms like depression and constipation is well-known [13,14].” with two new references: Baumgartner A, Press D, Simon D (2021) The relationship between olfactory dysfunction and constipation in early Parkinson’s disease. Mov Disord 36:781–782., Siderowf A, Jennings D, Eberly S, Oakes D, Hawkins KA, Ascherio A, Stern MB, Marek K; PARS Investigators. Impaired olfaction and other prodromal features in the Parkinson At-Risk Syndrome Study. Mov Disord. 2012 Mar;27(3):406-12.
Is RBD associated with olfactory disturbances in other synucleinopathies? The authors may provide references.
According to Reviewer suggestion Authors included the following sentence has been included on the text: “In fact, the assessment of olfactory impairment, and especially odor identification, may help to predict the onset of a Lewy body disease in patients with idiopathic RBD over a relatively short time period [27]. Furthermore, in RBD patients, anosmia predicts a higher short-term risk of transition to Lewy body disease, although it cannot distinguish between PD and Lewy body disease [28].” with two these new references:
Mahlknecht P, Iranzo A, Högl B, Frauscher B, Müller C, Santamaría J, Tolosa E, Serradell M, Mitterling T, Gschliesser V, Goebel G, Brugger F, Scherfler C, Poewe W, Seppi K; Sleep Innsbruck Barcelona Group. Olfactory dysfunction predicts early transition to a Lewy body disease in idiopathic RBD. Neurology. 2015 Feb 17;84(7):654-8.
Miyamoto T, Miyamoto M. Odor identification predicts the transition of patients with isolated RBD: A retrospective study. Ann Clin Transl Neurol. 2022 Aug;9(8):1177-1185.
Materials and Methods:
The methods have to be curtailed. Appropriate references of the validated scales and methods are sufficient. For eg. details of various domains for MoCA, UPSIT and details of RBDSQ may be omitted.
According to Reviewer suggestion, methods have been curtailed, while some details of various domains for MoCA, UPSIT and RBDSQ have been omitted.
Inclusion and exclusion criteria are vague.
According to Reviewer suggestion, inclusion and exclusion criteria of the study for PD patients were added.
Results:
Difference in gender for olfaction impairment is not clear. Language editing is required.
According to Reviewer suggestion, gender differences in olfactory impairment have been better explained with language editing.
Fig 1. Scatter plots may be improved by increasing the size.
According to Reviewer suggestion, the Scatter plots have been improved by increasing the size
Discussion:
RBD as the most important functional risk marker for prodromal PD is now known. Numerous studies have also evaluated the association between olfactory dysfunction and motor and non-motor symptoms in PD. Over here, the authors show that olfaction impairment and RBD to be strongly associated as their main finding and that anosmia is more frequent than RBD in PD patients. The gender difference is also highlighted.
There are certain concerns regarding the paper.
Firstly, questionnaires to screen for RBD are less specific and sensitive than video-PSG, the gold standard for testing RBD. However, it is understood that for such a large population of patients, PSG was not feasible.
We agree with Reviewer that questionnaires to screen for RBD are less specific and sensitive than video-PSG. In the manuscript we have acknowledged this as limitation” We also added that, due to this limitation, “this study should be considered as a pilot study.”
Secondly, exploring the relationship of olfactory impairment and RBD in less severe tremor-dominant PD compared to the severe PIGD motor subtype may give further insight into disease mechanisms. Longitudinal design of the study would have added more strength to the present findings
We strongly agree with the reviewer that exploring the relationship of olfactory impairment and RBD in less severe tremor-dominant PD compared to the severe PIGD motor subtype may give further insight into disease mechanisms. The following sentence has now included on the Discussion: “As a further limitation, we did not explore the relationship of olfactory impairment and RBD in less severe tremor-dominant PD compared to the severe PIGD motor sub-type. We retain that future investigations focused on this relationship may give further insight into disease mechanisms.”
Regarding the loss of a Longitudinal design as a limitation of this study, we have declared in the manuscript that “As a limitation, it should be noted that the study presented with a cross-sectional design, without investigating possible correlations over time and measure longitudinal changes.”
The last paragraph of the discussion can be eliminated.
According to Reviewer suggestion, the last paragraph of the discussion has been deleted.
Major language editing is needed in the revised paper.
The Manuscript has been revis
Reviewer 3 Report
Comments and Suggestions for Authors
This is a study elaborating on olfactory impairment as predictors of increased scores in REM behavior disorder in a population of Parkinson's disease patients. There are certain points which should be additionally adressed:
1. Regarding the impact on the olfactory structures of the neuroinflammatory processes in PD and neurodegenerative diseases, it would be valuable to elaborate on the neuroinflammatory pathogenesis of PD - Ref.
1. Platelet-to-lymphocyte ratio and neutrophil-tolymphocyte ratio may reflect differences in PD and MSA-P neuroinflammation patterns. Neurol Neurochir Pol. 2022;56(2):148-155. doi: 10.5603/PJNNS.a2022.0014. Epub 2022 Feb 4. PMID: 35118638.
2. Metabolic correlates of olfactory dysfunction in COVID-19 and Parkinson's disease (PD) do not overlap. Eur J Nucl Med Mol Imaging. 2022 May;49(6):1939-1950. doi: 10.1007/s00259-021-05666-9. Epub 2022 Jan 5. PMID: 34984501; PMCID: PMC8727173.
2. Due to the lack of video polysomnographic examination, this study should be considered as a pilot study.
3. Was the impact of pharmotherapy considered as a possible limitation?
4. The Table 1. should be corrected in the line of PD duration - information 2,7+-3,3 or 2,7 +-3,5 seems awkward.
5. A short elaboration on future perspectives should be added.
Author Response
This is a study elaborating on olfactory impairment as predictors of increased scores in REM behavior disorder in a population of Parkinson's disease patients. There are certain points which should be additionally addressed:
- Regarding the impact on the olfactory structures of the neuroinflammatory processes in PD and neurodegenerative diseases, it would be valuable to elaborate on the neuroinflammatory pathogenesis of PD - Ref.
- Platelet-to-lymphocyte ratio and neutrophil-tolymphocyte ratio may reflect differences in PD and MSA-P neuroinflammation patterns. Neurol Neurochir Pol. 2022;56(2):148-155. doi: 10.5603/PJNNS.a2022.0014. Epub 2022 Feb 4. PMID: 35118638.
- Metabolic correlates of olfactory dysfunction in COVID-19 and Parkinson's disease (PD) do not overlap. Eur J Nucl Med Mol Imaging. 2022 May;49(6):1939-1950. doi: 10.1007/s00259-021-05666-9. Epub 2022 Jan 5. PMID: 34984501; PMCID: PMC8727173.
We thank the Reviewer for the suggestion. We have added in the manuscript a sentence on this hypothesis and the following two new references:
- Madetko N, Migda B, Alster P, Turski P, Koziorowski D, Friedman A. Platelet-to-lymphocyte ratio and neutrophil-tolymphocyte ratio may reflect differences in PD and MSA-P neuroinflammation patterns. Neurol Neurochir Pol. 2022;56(2):148-155.
- Morbelli S, Chiola S, Donegani MI, Arnaldi D, Pardini M, Mancini R, Lanfranchi F, D'amico F, Bauckneht M, Miceli A, Biassoni E, Orso B, Barisione E, Benedetti L, Gianmario S, Nobili F. Metabolic correlates of olfactory dysfunction in COVID-19 and Parkinson's disease (PD) do not overlap. Eur J Nucl Med Mol Imaging. 2022 May;49(6):1939-1950.
- Due to the lack of video polysomnographic examination, this study should be considered as a pilot study.
According to Reviewer suggestion, we have added in the Discussion the following sentence: “In this context, the use of RBDSQ in this large population appears reasonable for the screening of this complex parasomnia and this study should be considered as a pilot study.”
- Was the impact of pharmotherapy considered as a possible limitation?
No, it was not. According to inclusion criteria of PPMI, PD patients enrolled were PD drug naïve. In this regard, we have added inclusion and exclusion criteria in Methods section
- The Table 1. should be corrected in the line of PD duration - information 2,7+-3,3 or 2,7 +-3,5 seems awkward.
We appreciated the suggestion of the Reviewer. Regarding this consideration, we have modified the tables in question, removing the symbol +-, and placing the standard deviation inside brackets. Even if the value of the standard deviation may appear inconsistent, it is correct because some patients, even if diagnosed recently, have indicated the first onset of symptoms as distant in time.
- A short elaboration on future perspectives should be added.
Authors in line to the Reviewer suggestion included the following sentence in the Conclusion section: “Evaluation of olfactory dysfunction and RBD may play a key role both in clinical practice and in the application of potentially neuroprotective treatments in PD patients.”
Round 2
Reviewer 3 Report
Comments and Suggestions for Authors
Authors implemented most of my suggestions, however regarding the topic of the manuscript, I believe that the manuscript should be extended by acknowledging the possible role of RBD in atypical parkinsonisms which are also affected by olfactory impairment - Ref.
Sleep disturbances in progressive supranuclear palsy syndrome (PSPS) and corticobasal syndrome (CBS). Neurol Neurochir Pol. 2023 Mar 17. doi: 10.5603/PJNNS.a2023.0019. Epub ahead of print. PMID: 36928793.
Author Response
Authors implemented most of my suggestions, however regarding the topic of the manuscript, I believe that the manuscript should be extended by acknowledging the possible role of RBD in atypical parkinsonisms which are also affected by olfactory impairment - Ref.
Sleep disturbances in progressive supranuclear palsy syndrome (PSPS) and corticobasal syndrome (CBS). Neurol Neurochir Pol. 2023 Mar 17. doi: 10.5603/PJNNS.a2023.0019. Epub ahead of print. PMID: 36928793.
We thank the reviewer for this suggestion. Thus, the manuscript has been extended by acknowledging the possible role of RBD in atypical parkinsonisms. For this reason, the following sentence has been added : "Furthermore, a recent review has highlighted that RBD, which have been primarily described in synucleinopathies, have also been found to be present in other atypical parkinsonisms such as progressive supranuclear palsy syndrome (PSPS) and corticobasal syndrome (CBS)"
Moreover, the following new reference has been added to the manuscript:
18. Alster P, Madetko-Alster N, Migda A, Migda B, Kutyłowski M, Królicki L, Friedman A. Sleep disturbances in progressive supranuclear palsy syndrome (PSPS) and corticobasal syndrome (CBS). Neurol Neurochir Pol. 2023 Mar 17.